Characterization of amphoteric bentonite-loaded magnetic biochar and its adsorption properties for Cu2+ and tetracycline

Deng Hongyan 1
He Haixia 1
Li Wenbin 1 lwb062@cwnu.edu.cn
http://orcid.org/0000-0002-8792-1851 Abbas Touqeer 2
Liu Zhifeng 3
1 College of Environmental Science and Engineering, China West Normal University , Nanchong , China
2 Key Laboratory of Environment Remediation and Ecological Health, Ministry of Education, Zhejiang University , Zhejiang , China
3 Qinba Mountains of Bio-Resource Collaborative Innovation Center of Southern Shaanxi Province , Hanzhong , China
Mortimer Monika
Electronic publication date: 2022 Mar 1
Publication date: 2022
Volume: 10
Electronic Location ID: e13030
Received 2021 Nov 1; Accepted 2022 Feb 8
Copyright: © 2022 Deng et al.
Copyright year: 2022
Copyright holder: Deng et al.
License: This is an open access article distributed under the terms of the Creative Commons Attribution License, which permits unrestricted use, distribution, reproduction and adaptation in any medium and for any purpose provided that it is properly attributed. For attribution, the original author(s), title, publication source (PeerJ) and either DOI or URL of the article must be cited.
License URL: https://creativecommons.org/licenses/by/4.0/

Keywords: Magnetized biochar, Bentonite, Dodecyl dimethyl betaine, Surface characteristics, Adsorption capacity

Funding: Fundamental Research Funds of China West Normal University 18B023 Tianfu Scholar Program of Sichuan Province 2020-17 Sichuan Province Science and Technology Support Program 2018JY0224 National Natural Science Foundation of P.R. China 41271244 This work was supported by the Fundamental Research Funds of China West Normal University (18B023), the Tianfu Scholar Program of Sichuan Province (2020-17), the Sichuan Province Science and Technology Support Program (2018JY0224) and the National Natural Science Foundation of P.R. China (41271244). The funders had no role in study design, data collection and analysis, decision to publish, or preparation of the manuscript.

==============================
To realize simultaneous adsorption of heavy metal and antibiotic pollutants by a BC-based recyclable material, Fe3O4 magnetic biochar (MBC) was prepared by co-precipitation method. Then different ratios of dodecyl dimethyl betaine (BS-12)-modified bentonite (BS-B) were loaded on the surfaces of biochar (BC) and MBC to prepare BS-B-loaded BC and MBC composites, called BS-B/BC and BS-B/MBC, respectively. The physicochemical and structural properties of the composites were characterized by scanning electron microscopy, Fourier transform infrared spectrometry, thermogravimetric analysis, specific surface area (SBET) analysis, and vibrating sample magnetometry, and the adsorption efficiencies of BS-B/BC and BS-B/MBC to Cu2+ and tetracycline (TC) were studied. The following results were obtained. (1) Compared with BS-B/BC, BS-B/MBC had decreased pH and cation exchange capacity (CEC) and increased SBET. The pH, CEC, and SBET of BS-B/BC and BS-B/MBC decreased with the increase in the BS-12 proportion of BS-B. The surface of BS-B/MBC became rough after Fe3O4 loading. (2) The residual rate of BS-B/MBC was higher than that of BS-B/BC after high-temperature combustion, and the residual rate decreased with the increase in the BS-12 proportion of BS-B. The 2D infrared spectra showed that Fe3O4 and BS-12 were modified on the surface of BS-B/MBC. MBC and BS-B/MBC had splendid magnetism and could be separated by external magnetic field. (3) Compared with unmagnetized ones, the adsorption effects of Cu2+ and TC on different BS-B/MBCs improved, and the average adsorption rate reached the largest value of 91.92% and 97.76%, respectively. Cu2+ and TC adsorptions were spontaneous, endothermic, and entropy-increasing processes. The pH and SBET of the material had a great influence on Cu2+ and TC adsorptions, respectively, than CEC.

Introduction

Heavy metal and antibiotics in livestock farm wastewater has become one of the focuses of research (Nyamukamba et al., 2019). The metabolites of heavy metals and antibiotics have strong persistence, difficult degradation, and easy accumulation; can exist in water and soil environment for a long time; and eventually threaten human health through the food chain (Khan et al., 2015; Min et al., 2018). Therefore, screening remediation materials with high adsorption capacity and good recycling performance for heavy metal and antibiotic have great importance for the pollution control of livestock and poultry breeding and the sustainable development of agriculture.

The remediation methods for water pollution include physical, chemical, and biological remediation (Mazurkiewicz et al., 2020; Zulfiqar et al., 2021; Kumar et al., 2021). Among which, material adsorption has become a hot spot in pollution remediation research because of its simple operation, low cost, and obvious effect (Fu et al., 2019; Isakovski et al., 2020; Kumar et al., 2021). Many repair materials, such as biochar (BC) (Ahmad et al., 2014; He et al., 2019), clay minerals (Beraa et al., 2016), and agricultural and forestry wastes (Garg et al., 2021), have been studied. The surface of BC contains a large number of negatively charged functional groups, which greatly adsorb heavy metals and organic pollutants in water environment (Mohan et al., 2014; Yang et al., 2016). The adsorption capacities of straw BC for Cd2+ and methylene blue are 30.19 and 46.60 mg/g, respectively (Park et al., 2016; Li et al., 2016a). Feng et al. (2020) found that the adsorption capacity of bamboo–willow BC for oxytetracycline and sulfaethoxazole are 11.98 and 10.12 mg/g, respectively. The adsorption isotherm models of BC for phenol and tannic acid are multilayer and monolayer, respectively (Lawal et al., 2021). BC materials have good adsorption effect on pollutants but are difficult to separate in aqueous solution. Some researchers loaded magnetic particles on the surface of BC by Fe3O4, and realized the separation of BC material in solution by magnetic force. Then, the pollutants absorbed on the magnetic biochar (MBC) are eluted for the recycling of the MBC (Yan et al., 2015; Wang et al., 2020). Studies have shown that MBC can form a weak magnetic field around them to improve the metabolic capacity of microorganisms, accelerate the decomposition of pollutants, and thus reduce the concentrations of pollutants (Kastner, Mani & Juneja, 2015; Kyung-Won et al., 2016).

Bentonite (B) has strong adsorption capacity, ion exchange capacity and expansibility as a good pollution remediation material (Said & Goda, 2021). Current research on organic B focuses on its modification using surfactants to enhance its adsorption and fixation of pollutants. Amphoteric surface modifiers have hydrophilic positive and negative charges and hydrophobic carbon chains, which can adsorb organic and heavy metal pollution, simultaneously. Therefore, using amphoteric surface modifiers to modify B can simultaneously improve the adsorption performance of B to organic matter and heavy metals (Li et al., 2016b). The adsorption amounts of Cd2+ and phenol on Lou soil modified by amphoteric surface modifier (BS-12) were 1.3–1.8 and 4.0–8.3 times higher than those on unmodified soil sample, respectively (Meng, Zhang & Wang, 2007). The magnetization method realizes the secondary utilization of BC but does not improve its adsorption capacity for pollutants (He et al., 2019; Wang et al., 2020). Therefore, loading amphoteric B on MBC can greatly improve the adsorption capacity of the MBC to various pollutants and realize the recycling of the material synchronously.

Alternanthera philoxeroides BC and BS-12-modified B (BS-B) were prepared by oxygen-limiting high-temperature pyrolysis and wet method, respectively, and MBC was prepared by co-precipitation method to verify the absorption effect of BS-B-loaded MBC (BS-B/MBC) on heavy metal and antibiotic pollution. Then, BS-B-loaded BC (BS-B/BC) and BS-B/MBC were prepared by separately loading BS-B on the surfaces of BC and MBC, and the properties and structure of the composite materials were characterized. In addition, the isothermal adsorption and thermodynamic characteristics of Cu2+ and tetracycline (TC) on the composite materials were studied to provide a reference for the application of BC-based composite materials in sewage treatment.

Materials and Methods

Materials

Dodecyl dimethyl betaine was used as the amphoteric modifier, which abbreviated as BS-12 (AR; produced by Tianjin Xingguang Reagent Factory, Tianjin City, China). The B used was sodium B, which purchased from Henan Xinyang Bentonite Produce Company and purified by washing method before use (Shah et al., 2018). The basic physicochemical properties of the purified B are: cation exchange capacity (CEC) is 1,000.33 mmol/kg, pH is 9.59, and total organic carbon (TOC) is 4.98 g/kg. FeCl3·6H2O, FeSO4·7H2O and NaOH were all purchased from Chengdu Kelong Chemical Reagent Factory, Chengdu City, Sichuan Province, China. TC was purchased from Sigma (St. Louis, MO, USA) and had a purity of 99.9%. Cu2+ solution was used as pollutant, and the solution was prepared by using CuSO4·5H2O (analytical reagent) purchased from Chengdu Kelon Chemical Reagent Factory. Figures 1A and 1B show the structural formula of BS-12 and TC, respectively.

Figure 1 Structural formulas of BS-12 (A) and TC (B).

BS-B was prepared by wet method (Li et al., 2016b). BS-12 solution was added into 10 g of purified B (the mass ratio between solution and soil was 10:1), then reacted for 6 h at 40 °C; Centrifuged at 4,800 r/min for 10 min, washed 3 times with deionized water (dH2O), dried, grinded and sieved through 60-mesh nylon sieve to obtain BS-B. The amount use of BS-12 modifier was defined by Eq. (1):

(1) W=m×CEC×M×10−6×RBS/CBS

where WBS stands for the mass (g) of BS-12; m stands for the mass (g) of bentonite; CEC stands for the cation exchange capacity of the bentonite (mmol/kg); MBS stands for the relative molar mass of BS-12 (g/mol); RBS stands for the modification ratio (50% or 100%) of BS-12; CBS stands for the content (mass fraction) of BS-12.

A. philoxeroides was washed with dH2O, dried to constant weight under 60 °C, grinded and sieved through a 200-mesh nylon sieve, and fired for 8 h under 400 °C by oxygen-limiting high-temperature pyrolysis to obtain BC. Co-precipitation method was used to prepare MBC (He et al., 2019). In this method, 20.00 g BC was dispersed in 2.0 L dH2O and stirred for 30 min. Under strictly anaerobic conditions, 0.4 M FeCl3·6H2O and 0.2 M FeSO4·7H2O were successively added to 60 °C water, fully stirred for 2 h, and heated to 75 °C. The pH was adjusted to 10 with 5 mol/L NaOH solution and the MBC was separated by magnets after continuous stirring for 1 h and natural cooling. MBC was obtained after washing several times with dH2O. MBC was dried at 60 °C and then passed through a 60-mesh sieve. A wet process was used to prepare BS-B/BC and BS-B/MBC. In this process, 100 g BC or MBC was slowly added to 1.0 L dH2O, and BS-B was added again. After stirring at 40 °C for 3 h, the samples were separated, washed by dH2O thrice and dried at 60 °C. Then, BS-B/BC and BS-B/MBC were dried at 60 °C for 12 h and passed through a 60-mesh sieve.

Experimental design

BS-B samples with BS-12 proportions of 0%, 50%, and 100% were prepared and named as B (bentonite only), 50BS-B, and 100BS-B, respectively. BC and MBC were used as the control (CK). BC, B/BC, 50BS-B/BC, 100BS-B/BC, MBC, B/MBC, 50BS-B/MBC, and 100BS-B/MBC (8 samples in total) were analyzed for the pH, CEC, and Brunaer–Emmett–Telle specific surface area (SBET), and characterized by scanning electron microscopy (SEM), thermogravimetry (TG), Fourier transform infrared (FT-IR) spectroscopy, and vibration sample magnetometry (VSM).

The pre-experiment of Cu2+ adsorption showed that the adsorption isotherm began to turn at 300–400 mg/L, so the Cu2+ concentration in isothermal adsorption experiment was set to 0, 20, 50, 100, 150, 200, 300, 400, and 500 mg/L for a total of nine concentration gradients. The concentration of tetracycline was set to 0, 2, 5, 10, 20, 40, 60, 80, and 100 mg/L nine concentration gradients by their pre-experiment. Three replicates were set for each treatment.

Experimental methods

pH value was determined using a HQ411D table pH meter (Hash Company, Vancouver, WA, USA; refer to the test method of soil sample, solid–liquid ratio was 1:5). CEC was determined by sodium acetate–ammonium acetate method. SBET was analyzed by by multi-point BET method using a V-Sorb2800P analyzer. SEM was performed using a Japanese Hitachi S-4800 scanning electron microscope. TG was performed using a STA449F3 synchronous thermal analyzer (NETZSCH) under the following conditions: temperature range 25–900 °C; sample quality, 10–15 mg; heating rate, 10 °C/min; N2 atmosphere. FT-IR analysis was performed on a Nicolet 5DX type Fourier transform infrared spectrometer, and the 2D FTIR spectra were analyzed by 2DShige software. Magnetic curves were determined by Lakeshore 665 VSM method.

Nine samples (0.5000 g) of each composite material were separately packed in 50 mL plastic centrifuge tubes, added with 20 mL of Cu2+ (TC) solutions under different concentration gradients, shaken at room temperature for 12 h (200 r/min), and centrifuged at 4,800 r/min for 15 min. The supernatant was collected to determine the Cu2+ (TC) concentration, and the actual adsorption amount of the test material was calculated by subtraction (Zhang et al., 2020; Zou et al., 2020). The experiments were carried out at 20 and 40 °C respectively for calculating the thermodynamic parameters of Cu2+ and TC adsorptions. The Cu2+ content was determined via flame atomic absorption spectrophotometry, and background absorption was corrected through the Zeeman effect. The TC concentration was determined by SP-2100 UV-VIS spectrophotometer at 365 nm. The above measurements were all inserted into standard solutions for analysis quality control.

Data processing

The equilibrium adsorption amount of Cu2+ and TC was calculated using Eq. (2):

(2) q=V×(C0−Ce)W0

where C0 (mmol/L) and Ce (mmol/L) are the initial and equilibrium concentrations of Cu2+ (or TC) in the solution, respectively. V (mL) is the volume of Cu2+ (or TC) solution added. W0 (g) is the weight of the tested material. q (mmol/kg) is the equilibrium adsorption amount of Cu2+ (or TC) on the tested material.

The adsorption rate AR (%) of Cu2+ and TC was calculated using Eq. (3):

(3) AR=100×(C0−Ce)C0

The Langmuir isotherm was selected on the basis of the adsorption isotherm trend and the isothermal equation Eq. (4) is as follows (Zhang et al., 2020):

(4) q=qmbCe1+bCe

where qm indicates the maximum adsorption amount of Cu2+ (or TC) on the different materials, mmol/kg; b represents the apparent equilibrium constant of the Cu2+ (or TC) adsorption, which can be used to measure the affinity of adsorption.

Parameter b in the Langmuir model is equivalent to the apparent adsorption constant of equilibrium constant, and the thermodynamic parameter calculated by b = K or Ka is called the apparent thermodynamic parameters; Eqs. (5)–(7) are as follows (Zhao et al., 2021):

(5) ΔG=−RTln⁡K

(6) ΔH=R(T1⋅T2T2−T1)⋅ln⁡(Ka,T2Ka,T1)

(7) ΔS=ΔH−ΔGT

where ∆G is the standard free energy change (kJ/mol), R is a constant (8.3145 J/mol/K), T is the adsorption temperature (T1 = 293.16 K, T2 = 313.6 K), ∆H is the enthalpy of adsorption process (kJ/mol), and ∆S is the entropy change of adsorption process (J/mol/K).

CurveExpert 1.4 fitting software was used in isothermal fitting, and SigmaPlot 10.0 software was adopted to improve data plotting. SPSS 16.0 statistical analysis software was used to process the experimental data for variance and correlation analysis (Li et al., 2020). SigmaPlot 10.0 software was adopted to improve data plotting. The data were expressed as the means with standard deviation, and different letters indicate significant differences among various amendments (Lopez et al., 2012). Analysis of variance was performed to determine the effects of amendments, followed by Tukey’s honestly significant difference test. Differences of p < 0.05 were considered significant (Ana et al., 2013).

Results

Basic physicochemical properties of the tested materials

Figure 2 shows the physicochemical characteristics of each test material. The pH and CEC of BC increased when B was loaded but decreased when BS-B was loaded, and the amplitude decreased with the increase in the BS-12 proportion of BS-B. SBET decreased when BC was loaded with B and BS-B, and further decreased with the increase in the modification ratio of BS-12 on BS-B. B/BC had a slight increase in pH and a higher increase in CEC, which may be caused by the similar pH and larger CEC of B than those of BC. BS-12 on BS-B can neutralize the alkalinity of B, and the long carbon chains of BS-12 can cover the surface of B and thus reduce the CEC of BS-B. Therefore, when BC was loaded with BS-B, the pH and CEC of BS-B/BC materials decreased with the increase in the modification ratio of BS-12. Moreover, when BC was loaded with BS-B, the interlayer or surface pores of BC were covered by BS-B, which increased the average particle size of BS-B/BC and resulted in the decrease in SBET. Compared with the unmagnetized ones, the magnetized materials had slightly reduced pH and CEC and remarkably increased SBET because the increase in Fe3O4 particles increased the roughness of a material’s surface and then increased its surface area (He et al., 2019).

Figure 2 Physical and chemical characteristics of the test materials.

(A–C) pH, CEC, and specific surface area, respectively. BC and MBC stand for original biochar and magnetic biochar, respectively. B/BC, 50BS-B/BC, and 50BS-B/BC were 0%, 50%, and 100% BS-12 modified bentonite loaded BC. B/MBC, 50BS-B/MBC, and 50BS-B/MBC represent 0%, 50%, and 100% BS-12 modified bentonite loaded MBC. The same as other figures. The different uppercase and lowercase letters indicate significant difference among treatments at p = 0.0 1 and p = 0.05 level, respectively.

SEM images of the test materials

The SEM image of the surface morphology of each test material is shown in Fig. 3. BC had a smooth surface and a regular pore structure. When BC surface was loaded with B, the surface of B/BC became rough, a few number of B particles were attached to the surface of BC, and some pores were also filled with B particles. When BC was loaded with 100BS-B, the surface smoothness of 100BS-B/BC increased, because the hydrophobic long carbon chain of BS-12 could form a layer of organic phase on the surface of 100BS-B, which could decreased the surface roughness (Zhang et al., 2020). When each composite material was loaded with Fe3O4, its surface became rough compared with the unmagnetized ones. A large number of Fe3O4 particles were attached to the surface of the material, and some pores were also filled with Fe3O4 particles. The results show that Fe3O4 had a large effect on the surface morphology of the magnetic material, and the iron oxide particles were dispersed on the carbon matrix, which increased the SBET (Xin et al., 2017).

Figure 3 SEM images of test materials.

(A–F) BC, B/BC, 100BS-B/BC, MBC, B/MBC, and 100BS-B/MBC, respectively.

TG analysis of the tested materials

The TG curves of the different materials are shown in Fig. 4. The test materials had different degrees of weight loss after high-temperature pyrolysis (900 °C), and the weight loss rates of the materials were greater after magnetization. The weight loss of the test materials is in the order: BC > 100BS-B/BC > 50BS-B/BC > B/BC, 100BS-B/MBC > 50BS-B/MBC > B/MBC > MBC. As the modification ratio of BS-12 increased, the weight loss rate of BS-B/BC and BS-B/MBC materials gradually increased, mainly because the surface active agent BS-12 was decomposable organic matter (He et al., 2019). Change in the TG curve showed three stages at the temperatures of 0–250 °C, 250–600 °C, and 600–900 °C, which represent the material water loss, organic matter decomposition, and crystal layer collapse, respectively.

Figure 4 TG curves of test materials.

(A–B) Unmagnetized and magnetized materials, respectively.

Table 1 shows the weight loss rate and differential thermogravimetric (DTG, mass change rate with time, %/min) peak temperature of each material in the three stages of the TG curve. The DTG peak temperatures of the test materials were kept between 67 °C and 81 °C in the water loss stage (Stage 1). The water loss rates of the test materials are in the order: 100BS-B/BC (8.26%) > B/BC (8.07%) > 50BS-B/BC (7.51%) > BC (6.68%) > 100BS-B/MBC (5.02%) > 50BS-B/MBC (4.74%) > MBC (4.59%) > B/MBC (4.17%). In the organic carbon decomposition stage (Stage 2), the DTG peak temperatures of the test materials was kept between 400 °C and 530 °C. The carbon loss rate of BC was only 11.6%, whereas the carbon loss rates of the other test materials were in the range of 15.1–23.48%. Carbon loss rate increased with the increase in the modification ratio of BS-12. Almost no weight loss was observed in all the tested materials in the crystal layer collapse stage (Stage 3), and the DTG peak temperatures were in the range of 550–900 °C. This part of weight loss change is only related to the composition and structure of the material itself and has nothing to do with the organic modification.

Table 1 TG (%) and DTG (%/min) curves of the test materials.

BC and MBC stand for original biochar and magnetic biochar, respectively. B/BC, 50BS-B/BC, and 50BS-B/BC were 0%, 50%, and 100% BS-12 modified bentonite loaded BC. B/MBC, 50BS-B/MBC, and 50BS-B/MBC represent 0%, 50%, and 100% BS-12 modified bentonite loaded MBC. The same information applies to the other tables.

Treatments	Dehydration Stage 1 (<250 °C)	Organic carbon decomposition Stage 2 (250–600 °C)	Crystal layer collapse Stage 3 (>600 °C)	Residual rate (%)	
Water loss rate (%)	Peak temperature (°C)	Carbon loss rate (%)	Peak temperature (°C)	Peak temperature (°C)	
BC	6.68	67.13	11.6	403.49	618.98	65.71	
B/BC	8.07	68.34	15.1	444.69	633.35	68.61	
50BS-B/BC	7.51	69.51	18.04	481.68	692.78	67.28	
100BS-B/BC	8.26	67.25	23.48	525.85	750.44	66.63	
MBC	4.59	68.16	15.96	436.72	640.11	75.19	
B/MBC	4.17	79.82	19.2	448.31	651.61	72.01	
50BS-B/MBC	4.74	80.67	20.07	451.87	652.43	71.18	
100BS-B/MBC	5.02	70.48	21.17	455.23	642.39	70.24	

FTIR and magnetic characteristics of the test materials

The 2D infrared spectra of the test materials are shown in Fig. 5A (synchronous correlation diagram) and Fig. 5B (asynchronous correlation diagram), The red and blue marks in the figures indicate positive and negative reactions, respectively. The test materials presented a strong characteristic absorption peak of Fe–O bond near 659–695 cm−1, which indicates that Fe3O4 was modified to the surface of BC materials. The peak in the vicinity of 990 cm−1 is the O–H surface modal vibration absorption on the carboxyl group, which is related to the carboxyl group and the amine group in the BS-12 molecule. This result indicates that BS-12 bound to MBC. A characteristic absorption peak containing the C=O functional group appeared at 1,600 cm−1, and the deformation vibration characteristic absorption peak of the C–H bond appeared near 2,920 cm−1. The absorption peak at 3,438 cm−1 was produced by the stretching vibration of the –OH bond. These results confirm that BS-B and Fe3O4 were loaded on the BC surface.

Figure 5 2D infrared spectra of the test materials.

(A–B) Synchronous and asynchronous correlation diagram, respectively.

The results of the hysteresis curve in Fig. 6 show that MBC and BS-B/MBC have good magnetic separation performance. The saturation magnetization of MBC was 43 emu/g. After BS-B loading, the magnetism of BS-B/MBC decreased slightly with the increase in BS-12 modification on BS-B. BS-B/MBC could still be separated by applying a magnetic field, which was of great importance for the recycling of BS-B/MBC materials.

Figure 6 Hysteresis curves and magnetic effects of the test materials.

Isothermal adsorption and thermodynamic parameters of Cu2+

The Cu2+ adsorption isotherms and adsorption rates of the test materials are shown in Fig. 7. The adsorption capacity of Cu2+ increased with the increase in equilibrium concentration, and the adsorption isotherm of the test materials to Cu2+ all conformed to the Langmuir model. The Cu2+ adsorption capacity of BC increased obviously after B loading. The Cu2+ adsorption capacity of BS-B/MBC increased with the increase in the modification ratio of BS-12 on BS-B. Compared with the unmagnetized ones, the magnetized materials had increased Cu2+ adsorption capacity.

Figure 7 Adsorption isotherm and adsorption rate of Cu2+.

(A–H) BC, B/BC, 50BS-B/BC, 100BS-B/BC, MBC, B/MBC, 50BS-B/MBC, and 100BS-B/MBC, respectively.

Table 2 shows the fitting parameters and thermodynamic parameters of Cu2+ adsorption by Langmuir model. The correlation coefficient reached an extremely significant level (p < 0.01). The maximum adsorption capacity (qm) of different materials changed from 135.07 mmol/kg to 322.00 mmol/kg. Compared with unmagnetized ones, the magnetized materials had higher qm for Cu2+. 100BS-B/MBC had the best adsorption effect on Cu2+; its qm reached 322.00 mmol/kg, and its average adsorption rate (ARe) was 91.92%. The values of the adsorption constant (b) of magnetized materials for Cu2+ were smaller than those of unmagnetized ones, which indicates that magnetization reduced the adsorption affinity of BC materials to Cu2+. This outcome might be due to the fact that Fe3O4 particles block the pores of BC materials, which results in the reduction of exchangeable cation sites on the surface. The b values of BS-B/BC and BS-B/MBC were larger than those of BC and MBC, which indicates that the adsorption affinity to Cu2+ became greater after BS-B loading. B had a larger CEC and could have an ion exchange reaction with more Cu2+, and B loading could promote Cu2+ adsorption. The value of b was the largest when BC was loaded with B but decreased with the increasing in BS-12 modification rate on BS-B when BC was loaded with BS-B. This result indicates that BS-12 formed an organic coating on the surface of BC and reduced its adsorption affinity for Cu2+.

Table 2 Langmuir fitting parameters and thermodynamic parameters for Cu2+ adsorption.

Treatments	Fitting parameters	Average adsorption rate (%)	Thermodynamic parameters	
Correlation coefficient/r	Standard deviation/S	qm (mmol/kg)	b	ΔG20 (kJ/mol)	ΔG40 (kJ/mol)	ΔH (kJ/mol)	ΔS (J/mol/K)	
BC	0.9923**	5.70	135.07	1.16	62.87	−17.13	−18.62	4.76	74.68	
B/BC	0.9948**	7.78	199.80	4.22	84.42	−20.25	−21.84	3.07	79.55	
50BS-B/BC	0.9905**	12.02	240.43	4.10	88.21	−20.15	−21.77	3.49	80.67	
100BS-B/BC	0.9897**	13.59	286.31	3.32	90.08	−19.70	−21.22	2.64	76.21	
MBC	0.9949**	5.20	157.18	1.06	66.29	−16.75	−18.43	7.89	84.03	
B/MBC	0.9978**	5.60	234.77	3.67	87.31	−19.76	−21.57	6.72	90.33	
50BS-B/MBC	0.9954**	8.89	264.49	3.92	89.93	−20.04	−21.61	3.00	78.60	
100BS-B/MBC	0.9968**	8.03	322.00	3.15	91.92	−19.52	−21.07	3.23	77.61	
Note:

** The correlation coefficient is significant at p = 0.01 level (r = 0.798 when the degree of freedom f = 7 and the level of significance p = 0.01).

At 20 and 40 °C, the Gibbs free energy (ΔG) values for the Cu2+ adsorption of each test material were less than 0, which indicates that the adsorption was a spontaneous reaction. The adsorption enthalpy change (ΔH) of each tested material to Cu2+ was greater than 0, which indicates that the adsorption was an endothermic reaction, and increasing temperature was conducive to the occurrence of adsorption. The entropy change (∆S) of each test material was greater than 0, which indicates that the disorder of the system in the Cu2+ adsorption process was increased by the tested materials.

Isothermal adsorption and thermodynamic parameters of TC

Figure 8 shows the adsorption isotherm and adsorption rate of TC on the tested materials. The TC adsorption capacities of the materials increased with the increase in equilibrium concentration and were in accordance with the Langmuir model. Table 3 shows the Langmuir model fitting parameters for TC adsorption. The fitting of the TC adsorption isotherms of the test materials reached a very significant correlation level (p < 0.01). The qm for TC adsorption changed from 118.60 to 602.83 mmol/kg, B/MBC had the best qm for TC adsorption, and 50BS-B/MBC had the largest ARe of 97.76%. Compared with the unmagnetized materials, the magnetized materials could reach the TC adsorption equilibrium quickly and had higher qm values for TC adsorption. After B and BS-B loading, the b values of the materials for TC adsorption were larger than those of BC and MBC. At 20 and 40 °C, all test materials for TC adsorption had ΔG < 0, which indicates that the adsorption was a spontaneous reaction; ΔH > 0, which indicates that the adsorption was an endothermic reaction; and ∆S > 0, which indicates that the disorder of the system increased in the TC adsorption process.

Figure 8 Adsorption isotherm and adsorption rate of TC.

(A–H) BC, B/BC, 50BS-B/BC, 100BS-B/BC, MBC, B/MBC, 50BS-B/MBC, and 100BS-B/MBC, respectively.

Table 3 Langmuir fitting parameters and thermodynamic parameters for TC adsorption.

Treatments	Fitting parameters	Average adsorption rate (%)	Thermodynamic parameters	
Correlation coefficient/r	Standard deviation/S	qm (mmol/kg)	b	ΔG20 (kJ/mol)	ΔG40 (kJ/mol)	ΔH (kJ/mol)	ΔS (J/mol/K)	
BC	0.9964**	0.70	336.59	3.53	89.54	−19.96	−21.34	0.41	69.47	
B/BC	0.9975**	0.58	152.82	10.20	92.72	−22.49	−24.09	0.86	79.66	
50BS-B/BC	0.9977**	0.56	118.60	14.30	93.83	−23.31	−24.96	0.92	82.63	
100BS-B/BC	0.9971**	0.62	155.11	9.16	92.20	−22.23	−23.82	0.97	79.15	
MBC	0.9945**	0.92	501.00	7.23	96.31	−21.67	−23.17	0.39	75.26	
B/MBC	0.9919**	1.10	602.83	7.82	96.77	−21.81	−23.35	0.76	76.98	
50BS-B/MBC	0.9955**	0.83	432.28	13.91	97.76	−23.24	−24.85	0.36	80.52	
100BS-B/MBC	0.9953**	0.84	425.92	12.32	97.60	−22.91	−24.57	1.20	82.27	
Note:

** The correlation coefficient is significant at p = 0.01 level (r = 0. 798 when the degree of freedom f = 7 and the level of significance p = 0.01).

Discussion

Correlation between adsorption and physicochemical properties

The qm and ARe values of each test material for Cu2+ and TC were linearly fitted to the physicochemical properties of the material, and the fitting results are shown in Table 4. CEC had positive correlations with the qm and ARe for Cu2+ and TC adsorptions; pH and SBET had positive correlations with the qm and ARe for TC adsorption but had negative correlations with the qm and ARe for Cu2+ adsorption; pH had a negative correlation with the ARe for TC adsorption. Moderate correlations between qm and pH, and between ARe and SBET were observed in Cu2+ adsorption (r > 0.5); and the other indexes maintained a low degree of correlation. qm and SBET in TC adsorption were moderately correlated, and the other indexes maintained a low degree of correlation. The results indicate that the pH and SBET of the material have a greater influence on Cu2+ and TC adsorptions respectively, than CEC.

Table 4 Correlation s between Cu2+ (TC) adsorption and physicochemical properties.

qm and ARe were maximum adsorption amount and average adsorption rate of Cu2+ (TC), respectively.

Adsorption parameters	Physicochemical properties	Regression equation	Correlation coefficients/r	Standard deviation/S	
Cu2+	q m	pH	pH = −0.01qm + 10.01	0.6039	0.44	
CEC	CEC = 0.09qm + 189.36	0.0642	80.79	
SBET	SBET = −0.06qm + 22.38	0.4696	5.74	
AR e	pH	pH = −2.22E + 10.62	0.4895	0.49	
CEC	CEC = 130.21E + 100.80	0.1980	79.36	
SBET	SBET = −27.25E + 33.72	0.5156	5.57	
TC	q m	pH	pH = 0.27qm + 14.55	0.2989	0.53	
CEC	CEC = 6.02qm + 79.86	0.0459	80.87	
SBET	SBET = 6.65qm − 130.72	0.6302	5.05	
AR e	pH	pH = −7.34E + 15.73	0.4238	0.50	
CEC	CEC = 56.66E + 154.80	0.0225	80.94	
SBET	SBET = 96.72E − 80.28	0.4785	5.71	

Adsorption difference of Cu2+ and TC on BS-B/MBC

Figure 9 shows the adsorption difference of Cu2+ and TC on BS-B/MBC. The SBET, pore structure, and functional groups of BC determined its adsorption capacity for Cu2+ and TC, but the adsorption capacity of original BC was limited. Solid–liquid separation could be realized under the action of external magnetic field by magnetizing the material with the magnetic medium. Fe3O4 loading increased the SBET and effective adsorption sites on the surface of BC. MBC and BS-B/MBC had good magnetic separation performances and could be separated by applying a magnetic field.

Figure 9 Adsorption differences of Cu2+ and TC on amphoteric-bentonite loaded magnetic-biochar.

B had strong adsorption capacity and ion exchange capacity, and amphoteric modification could remarkably improve its adsorption capacity for heavy metals and organic pollutants (Li et al., 2016c). Therefore, BS-B had high adsorption capacity for Cu2+ and TC. When MBC was loaded with BS-B, BS-B/MBC had a strong affinity for heavy metal ions and could effectively fix Cu2+, and which increases the adsorption capacity for Cu2+. The adsorption mechanism of Cu2+ comes from ion exchange, complexation, and electrostatic attraction by BC and from complexation and electrostatic attraction by BS-B (Zhang et al., 2020). TC has multiple ionizable functional groups, contains two groups of positive and negative charges, and has hydrophilicity (Khanday & Hameed, 2018). When the pH of the solution changed, the adsorption behavior of BS-B/MBC became more complicated. The relatively high surface area and large pore size of BS-B/MBC provided more active sites for TC molecules (Jang et al., 2018). The rapid adsorption stage dominated the adsorption process of TC. The adsorption mechanism of TC comes from ion exchange and hydrophobic combination by BC and from complexation and hydrophobic combination by BS-B (Zou et al., 2020). Moreover, the FTIR spectra of 100BS-B/MBC before and after Cu2+ and TC adsorptions were compared (Fig. 10). The results showed that the C=O and C–H bonds on the surface of the material shifted after Cu2+ adsorption, which indicates that C=O and C–H on the surface of 100BS-B/MBC were involved in the Cu2+ and TC adsorption processes. The appearance of O–H bonds on 100BS-B/MBC after TC adsorption showed that hydroxyl group participates in the TC adsorption process. Additionally, the movement of the peak to a higher wave number means that the energy required for vibration is lower, which indicates that the group is more stable.

Figure 10 FTIR spectra of 100BS-B/MBC before and after Cu2+ and TC adsorption.

100BS-B/MBC+Cu and 100BS-B/MBC+TC were 100BS-B/MBC have absorbed Cu2+ and TC, respectively.

Conclusion

After magnetization, BC had reduced pH and CEC and increased SBET. BC had a smooth surface, but the surface smoothness of BS-B/BC increased with the increase in BS-12 modification on BS-B. After Fe3O4 was loaded, a large number of Fe3O4 particles adhered to the surface of BS-B/MBC, and its surface became rough. Compared with unmagnetizatic ones, the magnetized materials had higher weight loss rates. FTIR and VSM analyses proved that BS-12 and Fe3O4 were modified to the surface of BS-B/MBC. MBC and BS-B/MC had good magnetic separation performances. BS-B/MBC had a good adsorption effects on Cu2+ and TC, and its ARe was more than 80%. Cu2+ and TC adsorptions were spontaneous, endothermic, and entropy-adding reactions. The pH and SBET of the material had a great influence on Cu2+ and TC adsorptions, respectively.

Supplemental Information

Supplemental Information 1 Raw data.

Click here for additional data file.

Additional Information and Declarations

Competing Interests

Author Contributions

Data Availability

The authors declare that they have no competing interests.

Hongyan Deng performed the experiments, analyzed the data, prepared figures and/or tables, authored or reviewed drafts of the paper, and approved the final draft.

Haixia He conceived and designed the experiments, performed the experiments, analyzed the data, prepared figures and/or tables, and approved the final draft.

Wenbin Li conceived and designed the experiments, analyzed the data, authored or reviewed drafts of the paper, and approved the final draft.

Touqeer Abbas conceived and designed the experiments, prepared figures and/or tables, and approved the final draft.

Zhifeng Liu conceived and designed the experiments, analyzed the data, authored or reviewed drafts of the paper, and approved the final draft.

The following information was supplied regarding data availability:

The raw measurements are available in the Supplemental File.

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
