# Peer review of "Characterization of amphoteric bentonite-loaded magnetic biochar and its adsorption properties for Cu2+ and tetracycline"

_PeerJ, doi:10.7717/peerj.13030_

## Round 0.1 · original submission · Major Revisions

Dear authors,

It is the opinion of three Reviewers that the manuscript needs substantial changes (including additional experimental data and rewriting of the manuscript) before it can be re-considered for publication. I agree with the Reviewers' opinion and ask you to consider the comments provided by the three expert Reviewers when improving your manuscript. Also, the language of the manuscript should be corrected. After you re-submit, the manuscript will be evaluated by Reviewers again.

Reviewer 1 ·

Basic reporting

no comment

Experimental design

no comment

Validity of the findings

no comment

Additional comments

In this manuscript, 8 biochar adsorbents were prepared and used to adsorb Cu and tetracycline. The topic is interesting. However, the mechanism analysis is not enough and the structure is not good. Therefore, "Major revision" is recommended. Specific comments are provided as follows:
1. The abstract is too long, it needs to be further simplified and highlight innovation. People care about the new mechanisms or extraordinary performance more.
2. The introduction was too short. Why the authors choose Cu and tetracycline as adsorbates? the authors were suggested to talk more about the Cu and tetracycline, including source, hazard, treatment method comparison, adsorption amount, etc.
3. Line 56, two spaces.
4. Line 86, how the amphoteric modified bentonites of “B (0%BS-B), 50%BS-B and 100%BS-B” were prepared? What’s 100%BS-B, no bentonite was used?
5. Section 1.1.1, please combine the sub-paragraphs into one paragraph.
6. How the pH value of solid sample was measured?
7. Space between value and unit was missing, such as line 101, 102. Double check the full text.
8. line 110-112, mL or ml? r/min or r min-1? Notice the color of text and commas, such as line 112-113, and line 141-145.
9. line 131-132, poor format.
10. line 166, bentonite (B), no need to abbreviate again.
11. The meaning of annotation in Figure 1 needs to be explained.
12. The resolution of SEM image is too low, so it needs to be updated. In addition, the ruler and other information in the image are not clear.
13. What is adsorption rate? How to calculate?
14. The author suggested that Fe3O4 particles block the pores of biochar, well then, why the surface area of MC was larger than C?
15. The governing mechanisms were unclear, which should be emphasized. XPS analysis needs to be added to support the adsorption mechanism.
16. Most of the cited references are out-of-date. It is suggested to cite the up-to-date literatures to highlight the significance and novelty of this research.
17. The English needs to be improved. The manuscript needs to be well organized. For example, combine the paragraphs with similar contents, and add the topic sentence at the beginning of each paragraph.
18. The conclusion needs to be more concise.

Reviewer 2 ·

Basic reporting

In this paper, amphoteric bentonite was loaded onto magnetic biochar in a novel direction and the adsorption effect of Cu2+ and tetracycline in water was studied. Authors used few characterization techniques to support their material properties, but fails to explain the data critically. Moreover, authors just focused on some preliminary studies, and they did not go in depth. Anyway, it could be re-considered for publication in this journal after the following revisions to improve its quality before publication.

1. The abstract was written like conclusion, please rewrite and state some key findings of your study.
2. The introduction is not well structured and less of clarity, especially paragraph 2.
3. The author should give a very clear introduction of BS-12, which is a major modifier in the experiment, like its amphoteric property.
4. There are some scientific, grammatical and typo errors in the manuscript that need to be re-checked and corrected more carefully. E.g., Line 47, “many repair materials studied more” should be “many repair materials were studied”; Line 56, two spaces between clay and using. Some sentences are too long. Others were highlighted in PDF.
5. All the figures are not clear, and the resolution are too low. I highly recommend that the authors should use different line types for the four lines in figure 3, not only use different color to distinguish them. Plus, what’s the meaning of Bb, Cc, Aa… above the columns in Figure 1?

Experimental design

6. The author should explain the experimental methods more detailed, like the heating rate and holding time for the preparation of biochar by pyrolysis. The meaning of the samples’ name should be clarified, such as 0%BS-B, 50%BS-B and 100%BS-B.

Validity of the findings

7. Line 156-158, a lower CEC value means better ion exchange ability? Magnetic modification and the addition of BS-B promoted reduced the CEC value while promoted the ion exchange?
8. The conclusion in section 3.1 are not that firm. If the BET of the samples is the key character to determine the adsorption effect of TC, then MC should show the best adsorption capacity because it has the highest surface area. There is no doubt that BET is very important for adsorption, but there must be some other properties also play a vital role, like CEC, although the correlation is poor. The authors also discussed the effect of CEC on adsorption in section 2.6. So I doubt the significance of the correlation analysis.
9. The MC100%BS-B has a lower BET and CEC, while has the highest qm for Cu2+, why?

Reviewer 3 ·

Basic reporting

The authors have made several biochar/bentonite composites, and also investigated the adsorption of Cu and tetracycline on these materials. In general, the English language of the whole manuscript is very unclear and ambiguous. There are many language defects throughout. The next most important issue is the inaccurate material naming. For example, the name of carbon-based amphoteric bentonite (CBS-B), should be a modified bentonite, not a biochar. So the abbreviation CBS-B seems to be ambiguous. Another problem is incomplete experimental data. For example, the authors calculated the adsorption thermodynamic parameters, but they did not conduct the thermodynamic adsorption experiments.

Experimental design

.Line 88, “and then magnetized by coprecipitation method loading Fe3O4 to prepare MC....” what is the detail method?
.Line 114, please provide the method reference.
Line 122, Is 12 hours enough? what is the adsorption equilibrium time?

Validity of the findings

Figure 1, What do the letters on the figures mean?
.Line 251-254, The data in Figure 7 and table 3 are inconsistent. The adsorption rate of 98.28% seems inaccurate, please check them carefully.

Additional comments

.The English language should be improved to ensure that an international audience can clearly understand your text. Some examples where the language could be improved include lines 21-22, 28-30, lines 96-102, etc.
.line 61-62, what is the scientific basis of this deduction?
.Your introduction needs more detail. The impact and novelty are not assessed.

---

## Round 0.2 · Minor Revisions

Please carefully double-check the manuscript and correct the language-related and methodological issues. In addition, Reviewer 1 has identified several mistakes in the manuscript which should be corrected before it can be further considered for publication.

Reviewer 1 ·

Basic reporting

The authors have solved most of the problems raised by the reviewers, however there are still many mistakes should be revised. Although the authors said they have proofread and redescribed the unclear description with the help of a native speaker, but the English still needs to be further improved. Major revision should be considered.

Experimental design

1. Line 110-114,“The pH, CEC, and Brunaer–Emmett–Telle specific surface area (SBET) of BC… were analyzed by scanning electron microscopy (SEM), thermogravimetry (TG), Fourier transform infrared (FT-IR) spectroscopy, and vibration sample magnetometry (VSM), respectively.” Wrong statement, these parameters cannot be tested by the instruments listed.
2. Line 125, the temperature range 25-1500 °C was not consist with the result of Fig. 4 TG curves of test materials.
3. Line 133-134, “The experiments were carried out at 20 and 40 ℃ respectively for calculating the thermodynamic parameters of Cu2+ and TC adsorptions” The common method to analyze thermodynamics is to require adsorption isotherm data at more than 3 temperatures, because the slope needs to be calculated during the analysis. However, the author only carried out adsorption experiments at 20 and 40 ℃, it is difficult to calculate the thermodynamic parameters accurately.

Validity of the findings

no comment

Additional comments

1. Line 13-16, such a long sentence.
2. Line 21, two spaces between surface and area.
3. Line 28, “lagest”, spelling mistake. The author should not make such elementary mistakes again in the revised edition.
4. Line 46, chargeed, wrong spelling.
5. Line 51, “The adsorption capacities of BC for phenol and tannic acid are multilayer and monolayer sorption, respectively”, grammatical error.
6. MBC or MBCs?
7. “Change in the TG curve showed three stages at the temperatures of 0–250, 250–600, and 600–900”, missing unit.
8. Line 227, Remove Spaces before punctuation marks.
9. Line 238, “indicats”, spelling mistake.

Reviewer 2 ·

Basic reporting

no comment

Experimental design

no comment

Validity of the findings

no comment

---

## Round 0.3 · accepted · Accept

Thank you for addressing the reviewers' comments and revising your manuscript. It can now be accepted for publication.